# Facilitators and Barriers of Medication Adherence Based on Beliefs of Persons with Bipolar Disorder: A Qualitative Study

**DOI:** 10.3390/ijerph19137633

**Published:** 2022-06-22

**Authors:** Jose Ángel Alcalá, Andrés Fontalba-Navas, Miguel Company-Morales, Samuel L. Romero-Guillena, Teófilo Gutiérrez-Higueras, Luis Gutiérrez-Rojas

**Affiliations:** 1Clinical Unit of Mental Health, Reina Sofia University Hospital, 14004 Cordoba, Spain; jaalcalapartera@comcordoba.com (J.Á.A.); teovillat@gmail.com (T.G.-H.); 2Northern Málaga Integrated Healthcare Area, Antequera Hospital, 29200 Antequera, Spain; andresfontalba@gmail.com; 3Department of Public Health and Psychiatry, University of Málaga, 29010 Málaga, Spain; 4Faculty of Nursing, Physiotherapy and Medicine, University of Almeria, 04120 Almeria, Spain; miguelcompanymorales@gmail.com; 5Nursing Service, Northern Almeria Health Management Area, Andalusian Health Service, Huercal Overa, 04600 Almeria, Spain; 6Clinical Unit of Mental Health, Virgen Macarena University Hospital, 41009 Seville, Spain; samuel.forqueta@gmail.com; 7Department of Psychiatry, School of Medicine, University of Granada, Tower A, Floor 9, 18071 Granada, Spain; 8CTS-549 Research Group, Institute of Neurosciences, University of Granada, 18011 Granada, Spain; 9Psychiatry Service, San Cecilio University Hospital, 18012 Granada, Spain

**Keywords:** bipolar disorder, acceptability, adherence, core beliefs, group intervention

## Abstract

One of the big challenges in treating individuals with bipolar disorder (BD) is nonadherence to medication. This is the principal factor associated with a worse prognosis or outcome of the disease. This study aimed to explore and analyze the individual perceptions that people with BD have about the positive and negative aspects when taking medication. A descriptive and interpretative study was carried out using the qualitative research paradigm with the use of the analytical technique of discourse analysis, extracting the data through the completion of focus groups. Participants’ speech was digitally audio-recorded in digital format. In order to complete the codification of the participants’ speech content, we relied on the qualitative data analysis (using the QRS NVivo 10 computer software). Thirty-six participants diagnosed with bipolar disorder took part in our study. In the participants’ speech concerning the main barriers to pharmacological treatment, three key topics were identified. Perceived facilitators were summarized in four factors. The main facilitators regarding the use of pharmacological treatment in individuals with BD were the ones related with the perceived need for treatment in the acute phase, the recognition of the illness, the shared clinical decision, and the causal biological attribution in the chronic phase. In terms of perceived barriers, social control was identified in both phases, adverse effects in the acute phase, and the absence of effective treatment in the chronic state.

## 1. Introduction

Bipolar disorder (BD) was significantly associated with a negative impact on the performance of work-related, leisure, and interpersonal activities [1]. These consequences usually involve social and workplace functioning, increasing the disability and worsening the quality of life for both mental and physical status [2]. It is important to identify the medication impact on quality of life, as it can lead to a reduction in the direct and indirect costs of the illness. The main therapeutic objective is to improve the clinical course, including a reduction in the frequency, seriousness, and consequences of manic, mixed, and depressive episodes, whereas the low adherence to treatment is the main modifiable factor associated with a greater recurrence of the illness [3].

As such, an appropriate pharmacological treatment may lessen the secondary problems associated with this illness. However, in an approximate interval of 20–66% of cases, the correct taking of the medicine is interrupted, with a median prevalence of 41% [4,5]; these differences can be explained, at least in part, because adherence has been defined and assessed differently in different studies [6].

The main factors involved with a correct adherence to the therapeutic recommendations are being a woman, higher level of insight, and being married; meanwhile, nonadherence to treatment is associated with being single, intensity of manic symptoms, negative attitude towards medication, substance abuse (heavy tobacco smoking and alcohol abuse), cognitive impairment, and comorbidity with personality disorder [7].

On the part of the people with BD, there are several subjective aspects which influence the decision about whether or not to follow the psychiatrist’s recommendations. These are fundamentally the perception of the illness, acceptance of the diagnosis, and attitudes towards the medication. Many of the patients have had negative subjective experiences that cause them to have erroneous beliefs about the effects that medication can produce on them [8,9]. These previous factors also vary in function depending on the life experience that the patients have; as they pass through the different stages of the illness, studies have highlighted that the factors associated with adherence may have more to do with erroneous and subjective experiences than with actual data [10]. For example, the fear of a possible side effect may prove to be more important than the actual side effects perceived in the decision to stop taking the medication [11]. In addition, beliefs regarding the illness and its treatment may also condition the attitudes and behavior of family members [12]. In this case, the qualitative methodology of research allows us to better understand BD and it helps to identify areas of intervention from the participant’s own point of view [13]. The aim of this study is to explore and analyze the individual perceptions that people with BD have about the positive and negative aspects when taking medication.

## 2. Materials and Methods

This work dealt with a descriptive and interpretative observational study under the qualitative paradigm of research, extracting the data through the completion of focus groups [14]. The process for the formation of ideas regarding people’s health was structured by a discursive phenomenon that carries personal and group practices and which covers different dimensions and social variables such as shared practices, values, beliefs, norms, and artifacts [15]. For this reason, we decided to carry out a study based on qualitative methodology using the analytical technique of discourse analysis.

### 2.1. Study Design

In our study, we considered the recommendations regarding qualitative research described by different authors [16,17] and, because of this, we tried to capture the meaning in the participants’ speech through a flexible and inductive way, from the specific to the general [18].

The main contribution obtained from a semi-structured interview results from the discourse of the participants. It is a method that seeks to understand how the individual views the social and the personal environments. This type of interview is relevant to the analysis of the meanings that social actors assign to their practices, and reveals the systems of values and norms on which these practices are grounded.

The interview had the following structure: in each meeting with the groups, the moderator welcomed and informed the participants, and made sure they had their consent to record their voices. Afterwards, the moderator asked key questions for the research with the aim of guiding the discussion (for example, what has been the main difficulty you have encountered during all these years due to your bipolar disorder? What is your opinion of the medication you have been prescribed to control the symptoms of the disease?). At the end of a discussion topic, the moderator gave a small summary of the group’s discussion and explained the agreements and disagreements. This allowed the discussion to end and the conversation to be guided to other topics. The meetings ended between 65 and 75 min from the beginning. The participation in these groups encouraged discussion and facilitated a flexible and open discourse, based on intersubjectivity and reflexivity [19]. Given the small number of participants in each group, the researchers encouraged all of them to participate in the discussion, which is one of the advantages of qualitative studies.

As an instrument, we chose the focus group because it provides numerous advantages: it promotes interaction, it offers direct information, it stimulates participation, it has a flexible and open nature, it presents the possibility to observe nonverbal components, and it favors intersubjectivity and reflexivity. For the design of focus groups, we followed the methodological recommendations of various authors [20,21,22,23,24]. The field notes and the in-depth observation regarding the study topic completed our methodological instruments, allowing for them to complete a “triangulation” among methods which verified or refuted the data of the focus groups [17].

### 2.2. Participants

The inclusion criteria were: a diagnosis of BD confirmed previously by their referred psychiatrist, the absence of clinical affective disorders, and no associated cognitive deterioration (COBRA scale comparable to healthy controls).

The sample of the health centers, as well as those participants in the research, were selected by means of an intentional or rational nonprobability sample [18]; thus, this was a convenience sample. An equal and non-discriminant selection of the sample was completed, being the one which best represented the study subject and not just the most accessible. A total of 10 voluntary participants were organized into each group. We designed four focus groups with 36 participants in order to select a large enough sample size to uncover a variety of opinions, but to limit the sample size at the point of saturation. Saturation occurs when adding more participants to the study does not result in obtaining additional perspectives or information.

The study was conducted in four health centers in different geographical places of Andalusia in the first quarter of 2017: two in urban environments, Granada and Seville, and two in rural environments, Palma del Río and Huércal-Overa, with the aim of achieving a socially-expansive representation. A fifth group was not organized because of information saturation [22,23,24]. The investigators had clinical experience in the follow-up of these individuals with BD, thus they selected the sample and included representative participants from each population.

The participants were selected for the sample by including representative patients from each population, sex, and origin equally, and including patients with enough years of evolution of the disease to know the main limitations and barriers that they have to face. The invitations and consent to participate were collected a week before the group meeting in order to cover any eventuality. Informed consent was obtained from all individual participants included in the study. The protocol of the study was approved by the Ethical Committee of Almería Centro.

The following diagnostic tools were used for the inclusion criteria:

(1) Diagnosis of BD complying with the diagnostic criteria of the DSM-5 [25], with at least 5 years of evolution and seeking parity to include the sex perspective. Representative people with BD were chosen among the general population. The place for the collection of data was the environment in which the regular follow-ups of these participants takes place: the community mental health units. They are the basic systems for specialized mental health care, constituting the first level of care and coordinating the rest of the attendance systems. They provide complete attention to patients in their population setting in outpatient or home regimes.

Intragroup homogeneity was sought after, in terms of age and place of origin (urban-rural surroundings). The urban-rural place of origin acted as a segmentation variable, by which two groups were planned regarding this variable (PROFILE 1: Urban, PROFILE 2: Rural). The differentiation criteria used was the size, placing the cut-off point by consensus in population centers with more or less than 50,000 inhabitants. The group was completed on 2 occasions for each profile, seeking information saturation in different provinces in order to identify possible cultural aspects related to the geographical location within Andalusia. Participants did not know each other.

(2) Preserved cognitive capacity was assessed by COBRA scale means.

The COBRA [26] is a 16-item questionnaire that measures BD patients’ perception of cognitive deficits in several areas, such as: executive functions, processing speed, working memory, learning and verbal memory, and attention/concentration. Responses are given on a 3-point Likert scale, ranging from 0 (never limited) to 3 (always limited). The total score is the result of the sum of the items. We used the Spanish version with a Cronbach’s alpha of 0.913. We did not include this scale to assess the patient’s cognitive status, but rather to ensure that all the participants did not have a cognitive impairment that would make it impossible for them to participate in the study.

(3) Clinical symptomatology: as a requisite for the participation in the study, the patient should be euthymic. For this, the Young Mania Rating Scale (YMRS) was used [27] for the assessment of manic symptoms and the Montgomery-Asberg Depression Rating Scale (MADRS) [28] for depressive symptoms. In order to access this study, the patients needed a score of <7 points in the YMRS and <6 points in the MADRS.

### 2.3. Categorization and Analysis of the Data

The speech of the participants was recorded in digital audio format. One of the researchers took notes and observed the nonverbal language of the participants, whereas another researcher performed moderation tasks. The intervention of each one of the participants was referenced by noting the minute of the intervention, the name of the participants, and the key words of their intervention. The analysis of nonverbal language took into account whether the participants showed concern, nervousness, verbosity, impatience, passivity, or irritability when expressing their opinions. The transcripts of the groups were completed by a professional transcriber. Some of the researchers (MCM and TGH) did not know the participants and analyzed the results independently. In order to complete the codification of the content of each participant’s speech, we relied on the QRS NVivo 10 computer software.

This methodology, which used triangulation, external validation, description in detail of the exact methods of data collection, analysis, and interpretation, supported the quality criteria of credibility, transferability, dependability, and confirmability.

Following the analytical schemes in qualitative research recommended by Atkinson and Hammersley [16], Ruíz [17], and Flick [18], we developed an analysis process which helped us to explain in further detail the reality of the participants’ discussions, describe the relationships among the discussion, and synthesize the data into an organized whole. In the codification work, we carried out synthesis and grouping processes in different analytical categories which dealt with the same discussion topic. The categories and the script were designed with the following scheme (see Table 1).

## 3. Results

Thirty-six participants took part in our study. Of the forty invited participants, four did not attend the group meeting because they were unable to attend. The ages ranged from 25 years of age to 60. Women participated at a slightly higher percentage in our research (*n* = 20, 55%). The average evolution time for the illness was 7.4 years (see Table 2).

### 3.1. Key Topic Categories

After the analysis of the participants’ speech (and the nonverbal language), different categories emerged which showed the beliefs which favored or hindered adherence to the pharmacological treatment for BD. These categories were grouped conceptually as barriers and facilitators (see Table 3). We analyzed the main facilitators and barriers associated with the highest psychological load from the nonverbal point of view. We defined facilitators as positive attitudes and beliefs about the change initiative and barriers as negative attitudes and beliefs towards the change initiative or the tasks involved in carrying out the change.

### 3.2. Perceived Barriers in Relation to the Pharmacological Treatment of Participants with Bipolar Disorder

In the participants’ speech, three key topics were shown (belief in the absence of effective treatment, adverse effects, and social control) as the main barriers to pharmacological treatment (see Table 4).

### 3.3. Perceived Facilitators Related to the Pharmacological Treatment of Participants with Bipolar Disorder

These facilitators are summarized in Table 5.

### 3.4. Relationship between Barriers and Facilitators in the Acute and Chronic Phase of Bipolar Disorder

Figure 1 summarizes the narrow relationships between the barriers and facilitators for the adherence of pharmacological treatment of BD within the acute and chronic phases of the illness. We have included this distinction (between the acute and chronic phases of the disease) because when analyzing the participants’ discourse, we observed that there was a great difference in the barriers and facilitators that prevented or improved adherence to treatment. Social control appears as a barrier to both periods.

When analyzing the barriers and facilitators reported by the participants, we found no differences in their clinical or sociodemographic variables.

## 4. Discussion

Although quantitative designs are commonly used in clinical research, some studies such as this require qualitative methods, and this methodology has proven to be useful in analyzing people with BD. After the qualitative analysis conducted in this study, the main barriers that appeared when undergoing pharmacological treatment were the fear of the adverse effects caused by the medicine, as well as the lack of confidence for an effective treatment.

In BD, social support, information received from the doctor, clinical depression, and the prior number of hospitalizations [29] have been described as factors related to the adherence of prophylactic treatment, as well as the results in our environment which related BD to cognitive deterioration [30]. Thus, our study provides factors that perhaps have not received enough attention from a quantitative point of view.

The recent study by Rosenblat [31], congruent to our work, identified the most important prognostic factors regarding tolerability and the effectiveness in frequency in a wide range of BD patents, similarly to how participants mentioned the most frequent adverse effects of treatment and the relationship with the acute phase of the disease.

Our work adds, regarding the knowledge of the topic, that the concept of the “effectiveness” is subjective, as it is a value that the person attributes to the treatment they are undergoing. If it strays from the supposition that there is no appropriate prophylactic treatment in the maintenance phase, the expectations of any attempt shall be low from the beginning, as well as the involvement in following it. From there, the need arises to explore this dimension before any prescription.

Qualitative methodology allows the exploration of new dimensions with individual factors, as well as group factors, appearing, which condition the attitude towards the treatment, much like the social control described by Foucault [32]. If the treatment is perceived as a coercive tool, the individual can take an initial fight position. However, the predominant discourse is learned helplessness, where the subject has a passive attitude in the face of the impossibility of changing the situation perceived as adverse.

Recognition of the illness (insight) and adherence to treatment, as well as the prognosis of the treatment, are widely described in schizophrenia and in BD [33]. Participants also included a perceived need for treatment, as simply knowing that you have a disorder may not be enough. In this case, we should remember that many individuals with BD do not receive effective pharmacological therapy in the early phases of the illness [34] and associate this delay as one of the main factors associated with the social and labor deterioration they have suffered.

The neurobiological substrate in BD is one of the most explained psychiatric disorders in the scientific literature, as is the effectiveness of lithium in the control of mania and the prevention of recurrences [35]. On the other hand, on a popular level, there is a strong belief in the causal attribution of BD to lithium deficiency and its internal imbalance, a belief that, despite being false, contributes greatly to alleviating the uncertainties surrounding the disease and provides a quick justification for the need for treatment.

Our study highlights the importance given by patients to take part in the decision about the treatment they should receive, especially in long-term follow-up (chronic phase). Others clinical studies have demonstrated the effectiveness of the decision shared by people who have BD [36,37] and other chronic psychotic disorders such as schizophrenia, showing effectiveness and improvement in the quality of decisions based on the increase in knowledge and participation, and observing a greater congruency with the values and preferences of the participant, as well as an increase in user satisfaction [37,38,39]. This shared decision making should be viewed as a dynamic process, where information is shared and reflected, and support is provided.

The social repercussion of the illness also emerged in the results of our study as a factor of great importance. Our participants pointed out that one of the main barriers they have had to face in living with this disease is that others consider that suffering from BD disqualifies them from working or take care of themselves. Qualitative studies noted that public stigma and discrimination were experienced from family, friends, and healthcare providers [40]; perhaps the shared clinical decision is particularly relevant to fight against this feeling.

Many participants believe that they need to take medicine with the aim of not being an inconvenience for the rest of the members in their social and family environment. They also have the perception of the treatment as a coercive measure and they would like to have a more active role in solving their BD. In this case, we should differentiate the manic phase of the bipolar illness, where the ability to make decisions is frequently changed and the low level of insight greatly impacts the ability to decide [41], from the remission phase of acute symptoms. In each stage of the illness, complete information about the possible treatments should be provided and the involvement of the participant in the choice of one medicine or another within the different options provided should be made easier. In addition, we may also improve the adherence and the confidence in the psychiatrist, key aspects in therapeutic effectiveness [7,42,43,44]. The fear of relapse is related to the fear of making wrong decisions about their care, something already pointed out in other recent studies with qualitative methodology [45]. There is also an important correlation between appropriate social and family support for the people with BD and the availability of this person to take mood-stabilizer medicines [46]. This approach should be taken into account, given that some authors [7] informed of nonadherence rates for these medicines between 20% and 66% [4].

In relation to the possible perceived facilitator effects, the concept of a shared clinical decision returns. In this case, it is necessary for the mental health professional to always remember, especially when attending to a person with BD, the evolution throughout the acute or chronic phase of the decision-making capacity, which must be differentiated by the existence of an opinion contrary to that of the perfectly reasoned expert [47].

The recognition of the illness and the perceived need for treatment appears in this case as factors related to perceived facilitators. In relation to the first factor, in the “self-regulation model” carried out by Leventhal [48], there was the perception of illness with the dimensions of identity, control, and evolution over time, including consequences and causes. Later, other authors [49] indicated that this previous model may be applied in mental illnesses such as psychosis, eating disorders, depression, and BD. Thus, the dimension of control is related to the search for effective strategies, for example, medicine that heals or improves the prognosis. In relation to the causal dimension, the worries or vital stressing factors, such as the existence of an imbalance in the brain, are usually frequently identified as etiological factors of bipolar illness on the part of the patients [50].

When we speak about the perceived need for treatment of BD, the causal biological attribution emerges as a facilitating element for the compliance of the treatment in our study. That is to say, it is easier for those individuals to consider the existence of a chemical problem in their brain because they frequently have the belief that their organism needs a substance or medicine to solve this change. In addition, these subjects usually have a greater adherence to the therapeutic measures for the illness. Additionally, in these cases, medication should be combined with psychotherapy, with the latter helping form a better self-knowledge of the patient, better self-care, and better understanding of the illness [51].

One of the principal strengths and novelties of this study is the use of qualitative research that offers more opportunities to gather important clues about any subject instead of being confined to a limited and often self-fulfilling perspective.

The main limiting factor in this study may be that the work was conducted in a regional environment by which some beliefs may be modified by cultural factors. In order to correct this limitation, an attempt has been made to achieve an appropriate homogeneity in terms of age, sex, preserved cognitive capacity, and place of origin as to apply the conclusions to different types of people suffering from BD. On the other hand, this possible limitation may be an advantage, as the results may be significant for other southern European countries with a very similar social and cultural configuration. Another limitation was that we do not use an objective method to evaluate the cognitive status of the participants.

To conclude this section, social control as a barrier appeared, once again, in the acute phase as well as in the chronic phase of the illness, with a view to correctly follow treatment for BD. In relation to the previous, the scientific evidence [52,53,54] supports the idea that strategies such as the correct taking of medicine, daily monitoring of mood changes, sleep and an appropriate diet, regular physical exercise, detecting the early signs of recurrence, and handling the difficulties of daily life may facilitate the appropriate functioning for our patients affected by BD at a family, social, and workplace level. In addition, some of these studies [55] place a great importance on the relationship between the correct acceptance of the personal impact that the BD causes in the person and the perceived improvement in quality of life [2]. It is very important that psychoeducation programs try to explain these advantages of medication and distance it from the stigmatizing concept of social control, reducing the number of relapses [56].

The positive association between treatment alliance and adherence in BD could be attributed to a number of intervening variables or mechanisms. An effective alliance results in less negative attitudes, a greater acceptance of illness, and the ability to tolerate medication side effects, eventually leading to improved adherence [57,58].

The assessment of the current situation in the adherence of the patient to the treatment and the identification of perceived barriers and facilitators of each patient can be used in planning an implementation and effect study, including the creation of an implementation plan to improve adherence.

## 5. Conclusions

In our qualitative study, we analyzed the factors that should be evaluated from the clinical point of view if physicians want to know what are the subjective reasons for the lack of adherence to pharmacological treatment. The barriers and facilitators that account for pharmacological nonadherence to treatment in persons with BD vary from patient to patient and may be patient-specific. As clinicians, we must know and explore what they are in order to develop an individualized strategy for each patient if we want to reduce the number of relapses and increase the quality of life of the people we treat every day.

## Figures and Tables

**Figure 1 ijerph-19-07633-f001:**
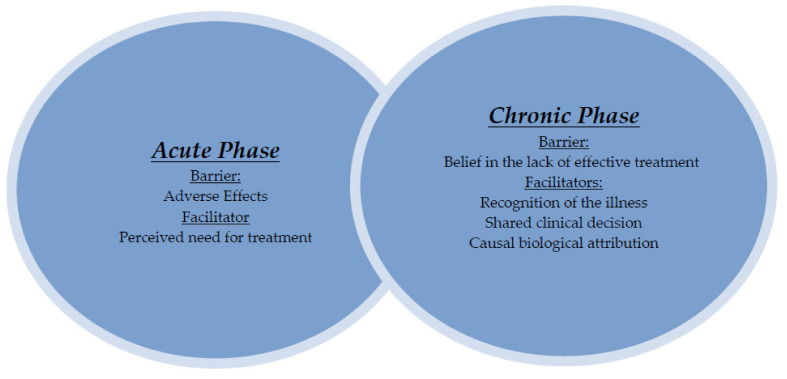
Barriers and facilitators in the adherence to treatment in the acute and chronic phases.

**Table 1 ijerph-19-07633-t001:** The categories and the script were designed with the following scheme.

Treatment	Acute phase
	Maintenance
Polarity	Mania
	Depression
Adherence	Barriers
	Facilitators

**Table 2 ijerph-19-07633-t002:** Sociodemographic and clinical characteristics of the sample.

*n*	36
% women	55%
Average age	40.2 years (sd 5.42)
Job	Employed 27.77%; unemployed 38.88%; retired 5.55%
Years of evolution of BD (average)	7.4 years (sd 1.04)
Average Young Scale	0.15 (sd 0.03)
Average MADRS Scale	2.36 (sd 1.12)
Average COBRA	6.5 (sd 5.06)

Sd: standard deviation.

**Table 3 ijerph-19-07633-t003:** Main facilitators and barriers regarding the use of pharmacological treatment in participants with bipolar disorder.

**FACILITATORS**	Recognition of the illness
	Perceived need for treatment
	Causal biological attribution
	Shared clinical decision
**BARRIERS**	Belief in the lack of effective treatment
	Adverse effects
	Social control

**Table 4 ijerph-19-07633-t004:** Barriers to the pharmacological treatment of bipolar disorder.

Category	Quotes
**Adverse effects**	Perceived need for treatment, “(…) and I’m destroyed, two pills and I’m destroyed, “bang!”, and I’m raring to go, and I don’t know what’s going on, I said: “Christ, what is this?” (…) you get vertigo, you feel bad. Sometimes, you don’t know how to -excuse me- wipe you arise or you can’t do it, you are embarrassed, people yell at you, you don’t know why… In other words, it’s crap, as you say: “Christ!”. It’s because they sedated you and remove the ability to react violently, obviously (…)”. “(…) some are to pick you up, others are to bring you down, others put you to sleep and others wake you up (…) and, in between all that, you get fatter, you lose your libido, your cholesterol goes up”.
**Belief in the lack of effective treatment**	“(…) it is very difficult for them to give us something that works”. “(…) there’s no magic bullet, not for them, not for us”. “I believe that if they knew how to fix it, they would’ve fixed it, but (…) they don’t know, the… the psychiatrists are taking stabs in the dark”. “They don’t have the tools. I don’t have the tools to come down or go up either”.
**Social control**	“It’s because we live in a society and, because of that, we don’t go without medicine; if we didn’t live in society, we wouldn’t take medicine because we wouldn’t bother anyone”. “(…) they forced me to take injections… and they told me “We can’t give you more”, that’s it, I didn’t even know what I had”.

**Table 5 ijerph-19-07633-t005:** Facilitators in the pharmacological treatment of bipolar disorder.

Category	Quotes
**Recognition of the illness**	“(…) I have to take medicine for my bipolar disorder, that’s it, I have a treatment, my illness has a name”. “I lowered the dosage myself because I’ve spent years more or less understanding this illness.” “(…) an illness that restricts us, the medicine we take has many effects that go against our health and, in my case, I’m type 1, mania is the one in charge”.
**Perceived need to treatment**	“There are times when I’ve gone to the doctor and I’ve told them: “Please, give me something, I can’t do this on my own” and then they give me something, still at risk. Now I have an anti-depressant and I’m somewhat ok and if they say they’re going to take me off it (…) I’ll tell them not to”. “(…) if you don’t have good medicine (…)”. “(…) you have to undergo treatment, if you don’t (…)”.
**Causal biological attribution**	“Of course I believe that the medicine has millions of benefits…what I want to say is that medicine is indispensable. They couldn’t take the medicine away from me. Because bipolarity has been studied…what happens in the brain is the movement of amounts of lithium. Therefore, lithium goes up and down”. “My mother also has a kind of disorder, like…she has depression, she has fibromyalgia and she has plenty of pills”. “With medicine, I’m ok for now, I also have rapid cycles”.
**Shared clinical decision**	“(…) the patient could also decide”. “Being sick doesn’t mean we can’t make decisions (…) but, as they think we’re crazy, they don’t let you make any decision”.

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
