# Peer review of "Facilitators and Barriers of Medication Adherence Based on Beliefs of Persons with Bipolar Disorder: A Qualitative Study"

_ijerph, 2022, doi:10.3390/ijerph19137633_

Round 1
Reviewer 1 Report
Dear authors,
These are my comments referring to your work:
- Introduction: It is necessary a better structure, a greater importance should be given to the notion of "adherence to treatment" (as a second paragraph). Before explaining the subjective aspects which influence the decision about to follow the psychiatrists' recommendations, I recommend to talk about the clinical factors contributing to a better/worse adherence to treatment.
- Material and methods: 1. what contains the semi-structured interview?2. explain the design of focus group.3. how many participants have each group? 4. what means saturation in this research?5. what means AFN?5. the inclusion criteria must be discussed early in this section.6. there is a paucity of the categories and the scripts.7. where are the mentions about the non-verbal language?
- Results: there is a need for a better systematization, the blue circles don't reflect anything. There are some statements better fits to "Discussion". There isn't a clear reflection of the tables in the content of the blue circles.
- Discussion: this section needs a better systematization, too. Every type of facilitator or barrier has to be discussed in relation with other studies.
- Conclusions: the conclusions present the results (unsubstantial expressed).
Author Response
These are my comments referring to your work:
- Introduction: It is necessary a better structure, a greater importance should be given to the notion of "adherence to treatment" (as a second paragraph). Before explaining the subjective aspects which influence the decision about to follow the psychiatrists' recommendations, I recommend to talk about the clinical factors contributing to a better/worse adherence to treatment.
Thank you very much. We have included in the second paragraph the principal’s factors associated with a better/worse adherence to treatment in BD patients.
- Material and methods:
2.1. what contains the semi-structured interview?
In the last version of the manuscript we have explained the contain of the interview in the Participants section. Now we have included this information in the study design section in order to improve the understanding of the methodology.
Now in the Study design section you can read the next paragraph:
The interview had the following structure: in each meeting with the groups, the moderator welcomed, informed the participants, and made sure they had their consent to record their voices. Afterwards, the moderator asked key questions for the research with the aim of guiding the discussion (for example, what has been the main difficulty you have encountered during all these years due to your bipolar disorder? What is your opinion of the medication you have been prescribed to control the symptoms of the disease?). At the end of a discussion topic, the moderator gave a small summary of the group’s discussion and explained the agreements and disagreements. This allowed to end the discussion and to guide the conversation among other topics. The meetings ended within 65 and 75 minutes from the beginning
2.2. explain the design of focus group.
We have added more details about the design of the groups.
Now in the Participants section you can read the next paragraph:
The participants were selected the sample including representative patients from each population, with equal sex and origin and with enough years of evolution of the disease to know the main limitations and barriers that patients diagnosed with this disease have to face.
2.3. how many participants have each group?
We have included this information in the manuscript.
Now in the Participants section you can read the next paragraph:
A total of 10 voluntary participants were organized for each group.
2.4. what means saturation in this research?
We have used qualitative methodology, in articles with similar approach is calculated that more than this number of participants do not give us more information. We have included more details about the meaning of saturation is.
Now in the Participants section you can read the next paragraph:
Saturation occurs when adding more participants to the study does not result in obtaining additional perspectives or information.
2.5. what means AFN?
The name of the moderator. It´s have been removed of the manuscript
2.6. the inclusion criteria must be discussed early in this section.
We have included the inclusion criteria in the first paragraph of this section.
2.7. there is a paucity of the categories and the scripts.
It is possible that we can included more categories in the study but we have limited time to evaluate each group (so as not to tire the participants). We chose all the principal items selected as topics of interest by the participants.
2.8. where are the mentions about the non-verbal language?
The qualitative methodology has the advantage of being able to analyze subjective aspects expressed by the participants. Those phrases that have been selected in the results tables are those that were associated with the highest psychological load from the nonverbal point of view.
We have included this detail in the results.
- Results: there is a need for a better systematization, the blue circles don't reflect anything. There are some statements better fits to "Discussion". There isn't a clear reflection of the tables in the content of the blue circles.
We agree with the reviewer so have moved all the comment about previous findings from Results section to Discussion section.
The figure 1 reflect the principal Barriers and Facilitators expressed subjectively for the participants after the evaluation. We think that it is very important to know what are the principal areas of interest of the patients in each phase (acute or chronic) of the illness but if the editor prefer to avoid it we do not have any problem to delete it.
- Discussion: this section needs a better systematization, too. Every type of facilitator or barrier has to be discussed in relation with other studies.
We have modified all the results discussed in relation with previous studies to de Discussion section. Recent studies have been added to the discussion
- Conclusions: the conclusions present the results (unsubstantial expressed).
We have re-written the conclusions section in order to avoid repeating the results described above.
Reviewer 2 Report
The present paper provides the results of a qualitative study on adherence to medication in Bipolar Disorder. It's worthy to notice how the authors have tried to focus on giving voice to people with mental health issues in a scientific paper which is laudable and not very usual. Nonetheless there are various improvements that should be done to the present work to clarify some obscure parts and recognize some flaws as limitations.
I have no major remarks on the Introduction Section
Regarding the Methods Section, it seems unusual and perplexing the use of the COBRA scale which the authors declare to use in order to assess cognition. The COBRA is innovative for its assessment of the subjective perception of patients of their cognitive deficits. It surely is an interesting instrument, but to exclude people with cognitive deficits structured interviews would have been more appropriate. The paper (Rosa et al., 2013)cited by the authors for example acknowledges its risk of bias and its defects in the assessment of cognition. The bias of the COBRA scale should be considered a limitation of the study in the choice of eligible participants.
In the Results section the authors should focus only on the exposition of what has arisen in the research, avoiding sentences that comment or contextualize the results which should be concentrated in the discussion section.
In the Discussion some opinions need reference or need to be clarified such as the following: "medicine is used in an iatrogenic way where there was no appropriate diagnosis." "The effectiveness of the decision shared by people who have BD". "it is an unacceptable error to identify mental illness with an inability to decide." Or some sentences seem to refer to results in the work but reference other articles (e.g. "They also have the perception of the treatment as a coercive measure and they would like to have a more active role in solving their BD"- the use of the pronoun They is confusing in the sentence because it seems like its referring to the specific BD cohort of the present article rather than referring to Ref n. 38 ).
In general, the word "patient" should be avoided and "participant" "person" or "individual are preferable, especially in an article which gives voice to people with BD and can be empowering.
Author Response
The present paper provides the results of a qualitative study on adherence to medication in Bipolar Disorder. It's worthy to notice how the authors have tried to focus on giving voice to people with mental health issues in a scientific paper which is laudable and not very usual. Nonetheless there are various improvements that should be done to the present work to clarify some obscure parts and recognize some flaws as limitations.
I have no major remarks on the Introduction Section
Regarding the Methods Section, it seems unusual and perplexing the use of the COBRA scale which the authors declare to use in order to assess cognition. The COBRA is innovative for its assessment of the subjective perception of patients of their cognitive deficits. It surely is an interesting instrument, but to exclude people with cognitive deficits structured interviews would have been more appropriate. The paper (Rosa et al., 2013) cited by the authors for example acknowledges its risk of bias and its defects in the assessment of cognition. The bias of the COBRA scale should be considered a limitation of the study in the choice of eligible participants.
Thank you for your interesting comment. We did not include COBRA scale to assess the cognition of the participants it was included only to ensure that participants did not have cognitive impairment that would make it impossible for them to take part in the study. We have included a clarification in both methods and limitations sections.
In the Results section the authors should focus only on the exposition of what has arisen in the research, avoiding sentences that comment or contextualize the results which should be concentrated in the discussion section.
Thank you very much for your comment. We have move these sentences to the Discussion section.
In the Discussion some opinions need reference or need to be clarified such as the following: "medicine is used in an iatrogenic way where there was no appropriate diagnosis." "The effectiveness of the decision shared by people who have BD". "it is an unacceptable error to identify mental illness with an inability to decide." Or some sentences seem to refer to results in the work but reference other articles (e.g. "They also have the perception of the treatment as a coercive measure and they would like to have a more active role in solving their BD"- the use of the pronoun They is confusing in the sentence because it seems like its referring to the specific BD cohort of the present article rather than referring to Ref n. 38).
Thank you very much. We have avoided the sentences that can be confusing. We have included recent studies to the Discussion section (all the new references are highlighted in yellow).
In general, the word "patient" should be avoided and "participant" "person" or "individual are preferable, especially in an article which gives voice to people with BD and can be empowering.
Thank you very much. We have changed the word patient for participant or person in the whole manuscript.
Round 2
Reviewer 1 Report
Dear authors,
The present form is much better written than the former one.
You respected all the indications I gave to you.
I am pleased with the actual form and I will recommend the publication of your paper.
This manuscript is a resubmission of an earlier submission. The following is a list of the peer review reports and author responses from that submission.
Round 1
Reviewer 1 Report
I thank the authors for the opportunity to review this interesting article. However, take into account the following recommendations and answer the questions raised:
• Specify the type of qualitative study carried out, explain the reason why you use this type of qualitative study. Is it an ethnographic or phenomenological study, does it use grounded theory, etc?
• When establishing the sample size, it takes 20 – 30 interviews as a reference (lines 95), why does it apply it to the discussion groups? They are different data collection tools. A focus group is not a group interview. Different data is collected and there is an interaction between the participants and researchers, it is not a good reference. Please justify this information.
• Detail the date on which the study was carried out.
• What quality criteria did you use in the study? How did you ensure credibility, transferability, dependency and confirmability?
• The main weakness in this manuscript is the results. In the presentation of the results, it refers to other studies, I am not sure why they do it, especially when there is a discussion section for it. They should only present the results of their own study.
Thanks a lot. All the best
Author Response
Thank you for your comments, Please see the attachment.

Reviewer 2 Report
Hello! This is an interesting article idea and something that could help clinicians in making decisions, however, I had a very hard time understanding much of what you wrote within the article due to grammar issues.
Here are some comments:
General
- Ensure that you are utilizing the past tense (participated, took part, etc) when writing as this is the standard for most scientific papers
- I am not sure what you mean when you say “facilitators” within your title or within the body of the paper
- You have several parts of your paper highlighted and/or in red font– I am not sure if this is simply something that you forgot to remove when submitting your paper; please remove the highlighting.
- Per submission requirements stated in the instructions for authors “In the text, reference numbers should be…placed before the punctuation; for example [1], [1–3] or [1,3].” Please correct this within the entirety of your paper
Abstract
- Lines 22 – 24 – sentence starting with “descriptive and interpretive observational…” this sentence is not grammatically correct; sentence starting with “the speech given…” also not grammatically correct. I can understand what you are trying to say with these two sentences, but it is not coming through with the way in which it is written
Introduction
- Line 37 - please correct grammar in the first two sentences sentence “…usually has negative consequences for the patients, their friends, and family members…
- Line 42 – please correct grammar “the therapeutic objective trying to achieve is the improvement…” this is very confusing, not sure what you are trying to say
- Lines 52 -53 – please correct grammar “There are another series of subjective aspects on the part of the patient, which influences the decision about to follow or not the psychiatrist’s recommendations.”
Material and Methods
- Specific Comments
- Lines 71 – 73 – this should be part of your discussion, this should not be in this particular portion (Although quantitative designs are commonly used in clinical research, some studies like this require qualitative method, and this methodology has proven to be useful in analyzing patients with BD)
- Lines 75 – 77 (Study design) – I am not sure what you are trying to say here
- Lines 90 – 99 (Participants) – I am not sure what you are trying to say here and am very confused about how you selected your participants
- Line 113 – who was the moderator? Were they a study author or an outside individual?
- Lines 159 – 161 – you discuss nonverbal language analysis; how was this done? Who conducted this?
- General Comments:
- Were participants given an incentive for their participation within focus groups or was it a completely voluntary thing?
- What where the questions that were asked of the participants in the focus groups?
- Where there any differences between the demographics or answers between the different focus groups, as they were from different geographical areas?
Results
- Specific Comments
- Lines 194 – 213 and lines 222 – 219 – this seems like you are discussing the results, which should be limited to the discussion section as the results is strictly for stating results not analyzing or comparing them
Author Response

(The authors gave the same response as above.)

Round 2
Reviewer 1 Report
I thank the authors for the opportunity to review the manuscript again, after their review I recommend:
• They still do not specify the type of qualitative study carried out, they focus on arguing the use of semi-structured interviews. These are a data collection tool, not a type of qualitative study. In the summary they give more information on this aspect, if you do not want to specify the type of study, put: "a descriptive qualitative study was carried out" and highlight the contributions of the qualitative studies compared to the quantitative ones in the study of the phenomena.
• Despite the fact that there are authors who defend the presentation of results together with their discussion, I think it is important that they present it separately, in order to better understand the manuscript.
Thank you very much. All the best
Author Response
They still do not specify the type of qualitative study carried out, they focus on arguing the use of semi-structured interviews. These are a data collection tool, not a type of qualitative study. In the summary they give more information on this aspect, if you do not want to specify the type of study, put: "a descriptive qualitative study was carried out" and highlight the contributions of the qualitative studies compared to the quantitative ones in the study of the phenomena.
In this study we use the analytical technique discourse analysis. We have included it in the manuscript (in the abstract section and in the Material and Methods section).
Despite the fact that there are authors who defend the presentation of results together with their discussion, I think it is important that they present it separately, in order to better understand the manuscript.
We agree with the reviewer. Now we have separated it and include this discussion explanation (included in the last version in the 3.3 section of the results) in the Discussion section of the actual version.
Thank you very much. All the best
Thank you very much for your nice comments.

Reviewer 2 Report
Thank you for your edits - my comments are below
- In your abstract you write "qualitative research paradigm" unclear what this means
- In response to your response, "We defined Barrier as a limit or boundary of any kind and Facilitators as thing that makes an action or process easy or easier." this still doesn't make any sense - a title should make it clear what the paper is about, and yours still is not clear even with this definition
- In the introduction, right after citation [12] you write "the aim of this study is to explore and analyses" this should be analyze
- In study design you write "The main contribution obtained from a semi-structured interview results from the density and level of complexity of the material obtained." Once again this is not clear and I am unsure what you are talking about
- In methods and materials, you write "The process for the formation of ideas regarding people’s health is structured by a discursive phenomenon that carry personal and group practices which cover different dimensions and social variables" What does this mean?
- Table 2 - you write "average years of evolution" what does this mean? What are you referring to?
- Table 3 - your fonts and formatting are not consistent within the table
- Table 4 and 5 - I like that you have various quotes which is always important in this kind of research, just not sure if there is any need to include participant number and group number in the table.
- After tables 4 and 5 you go into a discussion of other studies and start discussing your results - this should be in the discussion section not the results section
Author Response
Thank you for your edits - my comments are below
- In your abstract you write "qualitative research paradigm" unclear what this means
We have included in the abstract the subtype of qualitative analysis used in this study.
- In response to your response, "We defined Barrier as a limit or boundary of any kind and Facilitators as thing that makes an action or process easy or easier." this still doesn't make any sense - a title should make it clear what the paper is about, and yours still is not clear even with this definition.
We appreciate your comment. In order to improve the understanding, we have changed the definition.
Facilitators: positive attitudes and beliefs about the change initiative.
Barriers: negative attitudes and beliefs towards the change initiative or the tasks involved in carrying out the change.
Please let us know if the new definition is more clear.
- In the introduction, right after citation [12] you write "the aim of this study is to explore and analyses" this should be analyze
Thanks. We have change this word.
- In study design you write "The main contribution obtained from a semi-structured interview results from the density and level of complexity of the material obtained." Once again this is not clear and I am unsure what you are talking about.
We refer to the discourse of the participants; we have changed it in the test in order to improve the understanding of the text.
- In methods and materials, you write "The process for the formation of ideas regarding people’s health is structured by a discursive phenomenon that carry personal and group practices which cover different dimensions and social variables" What does this mean?
It is referred to the group’s shared practices, values, beliefs, norms and artifacts. We have included this explanation in the manuscript.
- Table 2 - you write "average years of evolution" what does this mean? What are you referring to?
It is referred to years of evolution of Bipolar Disorder, we have change it in the Table.
- Table 3 - your fonts and formatting are not consistent within the table
We have adapted it to the same fonts and formatting.
- Table 4 and 5 - I like that you have various quotes which is always important in this kind of research, just not sure if there is any need to include participant number and group number in the table.
We have avoided the number of participant and group.
- After tables 4 and 5 you go into a discussion of other studies and start discussing your results - this should be in the discussion section not the results section
We agree with the reviewer. Now we have separated it and include this discussion explanation (included in the last version in the 3.3 section of the results) in the Discussion section of the actual version.
